

# The most similar predictor - on selecting measurement locations for wind resource assessment

Andreas Bechmann[1], Juan Pablo M. Leon[1], Bjarke T. Olsen[1], and Yavor V. Hristov[2]

[1]DTU Wind Energy, Denmark
[2]Vestas Wind Systems A/S, Denmark

**Correspondence:** Andreas Bechmann (andh@dtu.dk)

**Abstract.** We present the "most similar"-method for conducting wind resource assessments with multiple wind measurements and for the optimal design of measurement campaigns.

Wind resource assessment is generally done by extrapolating a measured and long-term corrected wind climate at one location to a new location using a flow model. If several measurement locations are available, standard industry practice is to make a weighted average of all the predictions using inverse-distance weighting. The "most similar"-method challenges this practice. Instead of weighting several predictions, the method only selects the measurement location evaluated "most similar".

We validate the new approach by comparing against measurements from 185 met masts from 40 wind farm sites and show improvements compared to inverse-distance weighting. Compared to using the closest measurement location, the error of power density predictions is reduced by 13 % using inverse-distance weighting and 34 % using the "most similar"-method.

## 1 Introduction

### 1.1 The representativeness radius

When assessing the energy potential of a new wind farm, a crucial step is to predict the mean wind climate at each wind turbine position. The conventional approach for predicting the wind climate is to erect a meteorological mast (met mast) at a nearby location and extrapolate the measured wind climate to every wind turbine position using a micro-scale flow model (MEASNET, 2016). The flow model estimates how the surrounding topography perturbs the wind and can thereby predict the wind climate at each wind turbine position. The distance between the measurement and prediction location is traditionally used as an indicator of model uncertainty, and measurement campaigns are often planned to minimize the extrapolation distance. MEASNET (2016) defines a "representativeness radius" as the distance from a met mast to the furthest location that can be extrapolated with tolerable uncertainty. The representativeness radius depends on the complexity of the terrain, where complex terrain is characterized by having terrain slopes greater than 0.3. Figure 1 illustrates two met masts located in complex terrain. Following the recommendations of MEASNET (2016) the measurements taken at in the valley, $U_{01}$, are preferable for predicting $U_P$ as the distance is small.





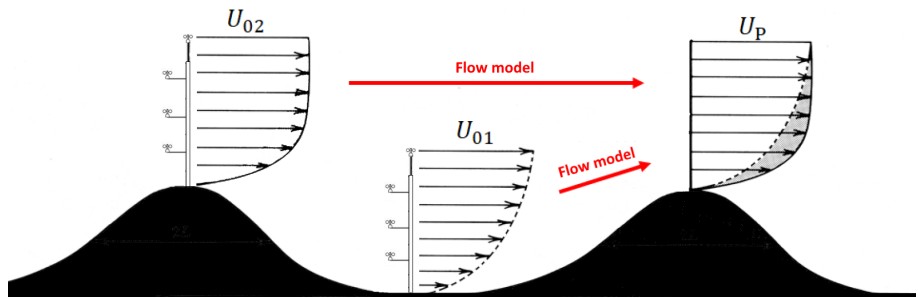

**Figure 1.** The figure illustrates how a measured wind climate can be extrapolated to a prediction site using a flow model. Following the "representativeness radius", the model error is reduced when the distance between met mast and prediction location is minimized; accordingly met mast 1 ($U_{01}$) is the preferred predictor. Following "the similarity principle" model errors are reduced when the wind conditions at the met mast and the predicted location are similar; accordingly met mast 2 ($U_{02}$) is preferable despite its more distant location

## 1.2 The similarity principle

According to Landberg et al. (2003), errors related to flow modelling are minimized when the predictor site (the met mast location) and the predicted site (the wind turbine location) are as "similar" as possible regarding factors like regional wind climate, roughness, orography and obstacles. Landberg et al. (2003) refers to this this as the "similarity principle". The underlying assumption is that no matter how advanced a flow model, it always produces errors, and they scale with the forcing applied. Figure 1 illustrates the similarity principle. The wind conditions in the valley, $U_{01}$, differ substantially from the conditions at the prediction site, $U_P$, and a better predictor (according to the similarity principle) is, therefore, the hilltop met mast, $U_{02}$, despite its distant location.

Experienced wind and site engineers determine suitable locations for met masts based on the representativeness radius and their judgment. They intuitively understand that distance is not the only parameter that should be evaluated; however, since no algorithmic method for similarity exists, the industry standard is to minimize the extrapolation distance.

## 1.3 The most similar predictor

Instead of using distance as the parameter representing similarity, we propose to use the directional-averaged speedup uncertainty, $\sigma_S$:

$$\sigma_S^2 = \sum_{j=1}^{N_\theta} f_j \sigma_{Sj}^2 \tag{1}$$

$$\sigma_{Sj}^2 = \left( \lambda \left[ 1 - exp\left( \frac{-d}{L_1} \right) \right] \right)^2 + \left| \frac{S_j - 1}{S_j + 1} \right|^2 \tag{2}$$





where $f_j$ is the wind direction frequency, $N_\theta$ is the number of wind direction sectors, $d$ is the extrapolation distance, $L_1 = 1\,\mathrm{km}$ and $\lambda = 0.1$ are empirically calibrated constants (Clerc et al., 2012), $S_j = U_{Pj}/U_{0j}$ is the speedup where $U_0$ and $U_p$ are the measured and predicted mean wind speed respectively.

$\sigma_S$ combines into one expression, the uncertainty associated with both extrapolation distance and speedup. Small values of $\sigma_S$ signify that predictor and prediction conditions are "similar". The inclusion of speedup in the expression ensures that

any difference between predictor and prediction site included in the flow model is evaluated (roughness, orography, obstacles, measurement height, etc.). The speedup uncertainty is, therefore, expected to be a more accurate "measure-of-similarity" than distance, and we define the "most similar" predictor to be the location with the smallest $\sigma_S$-value.

### 1.4 Multi-mast strategies

For large wind farms with several met masts, a standard practice when making predictions is to either select the closest available

predictor or to make a weighted average of multiple predictions using "inverse distance weighting". As an example, the inverse distance weighted mean wind speed, $U_P$, can be determined as,

$$U_P = \frac{\sum_{i=1}^{N_M} W_i U_{Pi}}{\sum_{i=1}^{N_M} W_i} \tag{3}$$

where $U_{Pi}$ is the predicted mean wind speed, $W_i = d_i^{-2}$ is the predictor weight, $d_i^{-2}$ is the extrapolation distance, and $N_M$ is the number of predictors on a particular site. The underlying reason for using inverse distance weighting is that the standard

error of a weighted mean decrease with the number of independent predictions. However, for this to be valid, the predictions are assumed independent (model errors should be random), and extrapolation distance is assumed to be the parameter that correlates the strongest with model error.

This paper aims to show that inverse distance weighting is not the optimal way of performing resource assessments. Instead, we suggest to follow the similarity principle and determine the "most similar" mast location (not the nearest mast). To validate

this approach, we conduct a large number of predictions, test the different multi-mast strategies and compare the results to measurements. We test the following multi-mast strategies:

**Closest predictor:**       $W_i = \begin{cases} 1 & \text{if } d_i = \min(d) \\ 0 & \text{otherwise} \end{cases}$  (4)

**Inv. dist. weighting:**       $W_i = d_i^{-2}$  (5)

**Most similar predictor:** $W_i = \begin{cases} 1 & \text{if } \sigma_{Si} = \min(\sigma_S) \\ 0 & \text{otherwise} \end{cases}$  (6)



## 2   Validation Method

To validate the usefulness of the "most similar" method, we compare the performance of the three multi-mast strategies against measurements. Sites with at least three met masts are used in the comparison. The multi-mast sites allow for predictions with multiple predictors and provide an objective way to evaluate the strategies. The validation method consists of three steps described in detail in the following sections:

1. **Measurements:** Preparation of the wind data for the flow model.

2. **Flow model:** Prediction of all mast locations using every other mast as a predictor.

3. **Wind statistics:** Calculation of mean wind speed and power density using each multi-mast strategy.

The measurements and model setup are identical for each multi-mast strategy; only the mast weights, $W_i$, used for calculating the wind statistics are different to allow for a simple and objective evaluation.

### 2.1   Measurements

A dataset has been collected to validate the multi-mast strategies. The dataset consists of measurements from sites with three or more met masts. By having at least three masts, each mast location can be predicted using at least two predictors, and results can be compared against the measurements taken at the predicted mast.

The dataset consists of wind speed and wind direction measured from the top anemometer of each met mast, already screened and long-term corrected by wind power developers. The data is grouped into 36 10-degree wind direction sectors, and have Weibull distributions fitted to the wind speed histogram. Wind statistics from a total of 210 met masts were provided for the study. The only additional screening that has been conducted for this study is the removal of four sites (25 met masts) from the dataset. These sites were removed since mast-to-mast predictions of wind speed, and power density led to substantial errors ($> 3\sigma$) for four of the met masts. We did not investigate the reason for the significant errors but removed the sites from the investigation. Table 1 shows a summary of the sites and met masts used. As seen, the screened data consist of 185 met masts distributed over 40 wind turbine sites.

The data comes from met masts located near potential wind energy installations, and the 185 mast locations represent varied site conditions from all over the world. Figure 2 illustrates the complexity of the sites using the RIX (left) and $\Delta$RIX (right) measure (Bowen and Mortensen, 2004). The mean RIX value is 7.0%, which can be considered moderately complex terrain. Compared to the closest neighbouring mast, 84 masts have $|\Delta$RIX$|$ values of less than 1% and 164 masts have $|\Delta$RIX$|$ below 5%. Most of the met masts are therefore placed in similar, well-exposed hills and ridges.

Figure 3 shows the height of the masts (right) and the distance to their closest neighbour (left). The distance is up to $10\,\mathrm{km}$, but on average the distance is $3.1\,\mathrm{km}$. As seen, the masts do not have the same height but vary from 20 meters to 120 meters, and half of the sites have height variations within them. These sites often have combinations of short and tall masts e.g. of 40/80 meters, 60/80 meters or 60/100 meter masts. The wind statistics have not been corrected or "sheared up" to unify the heights differences. Instead, the flow model has been used directly to estimate the wind statistics at the prediction height using



**Table 1.** Number of met masts used for the study.

| Masts per site | Sites | Masts | Cumulative sum |
|---|---|---|---|
| 25 | 1 | 25 | 25 |
| 7 | 4 | 28 | 53 |
| 6 | 3 | 18 | 71 |
| 5 | 5 | 25 | 96 |
| 4 | 8 | 32 | 128 |
| 3 | 19 | 57 | 185 |

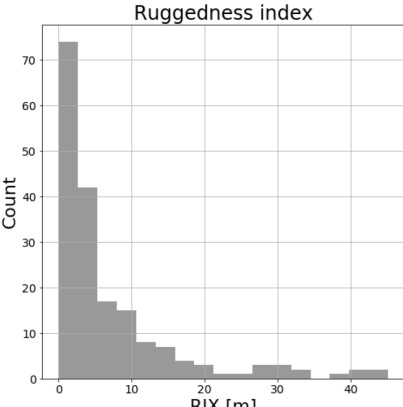
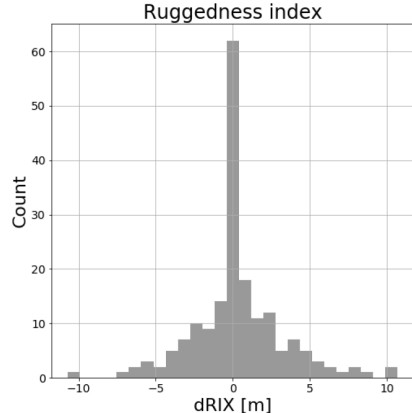

**Figure 2.** RIX and $|\Delta\mathrm{RIX}|$ for all 185 met. masts. $|\Delta\mathrm{RIX}|$ is calculated using the closest neighbouring mast.

the statistics at the predictor height. The "most similar" method have an advantage compared to the other multi-mast strategies for sites with different mast heights as there will be a speedup between masts of different heights and the will therefore not be valued as similar. In the current analysis, this is however of minor importance as the results, particular in Section 3.2, depend

mainly on a single site with 25 met masts that are all 80 meters tall.

The dataset (Table 1) allows for a total of 185 possible predictions with two or more predictor masts. The results section analyse how the different multi-mast strategies perform with this specific mix of multi-mast sites. Also, other combinations of multi-mast sites from the same data-pool have been made, to analyse how the multi-mast strategies perform on sites with a specific number of predictor masts. As an illustration of how other multi-mast combinations can be made from the same data,

we can imagine that a single site with 3 met masts can also be viewed as 3 different sites with 2 met masts. Table 2 shows the many combinations of sites possible with 1, 2, 3, 4, 5 or 6 predictor masts. As seen, it is possible to make more than 33



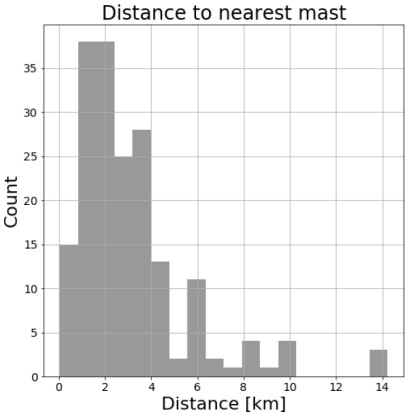
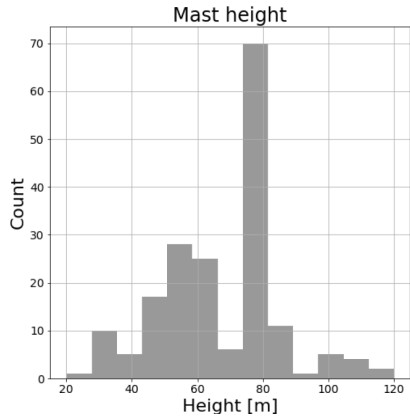

**Figure 3.** Mast height and distance to the closest neighbouring mast for all 185 met. masts.

**Table 2.** The table illustrates the number of predictions possible for combinations of sites with specific number of predictors.

| Predictors | 1 | 2 | 3 | 4 | 5 | 6 |
|---|---|---|---|---|---|---|
| Mast combinations | 1168 | 7803 | 51472 | 266185 | 1062786 | 33644928 |

million different predictions with 6 predictor masts. The many combinations are primarily possible due to the site that has 25 met masts. The results section show how the multi-mast strategies perform with these combinations of multi-mast sites.

## 2.2 Flow model

The flow model WAsP 12.3 (Troen and Petersen, 1989) has been used to make all wind climate predictions in this study. Since every mast location needs to be predicted using every possible predictor mast, there is a total of 1168 mast-to-mast predictions. A database and an automatic model script (PyWAsP) was made to conduct all the simulations, and every simulation was run with the standard WAsP 12.3 parameters. The topography maps and wind measurements needed were provided to the authors and used directly without conducting any corrections. Also, no $\Delta$RIX- or other corrections have been made to the model

results, and no tuning of model parameters to improve results was performed.

     This study represents a statistical analysis of multi-mast strategies. All simulations are based on identical model setups and using the digitised maps, mast locations, anemometer heights and wind data provided to the authors; only the method used to calculate the wind statistics is different between the 3 multi-mast strategies.



## 2.3  Wind statistics

Wind statistics are calculated at each prediction location to validate the performance of the multi-mast strategies. Specifically, the predicted mean wind speed, $U_P$ (ms$^{-1}$), and mean power density, $E_P$ (Wm$^{-2}$), is compared to the measured values ($U_0$ and $E_0$). Since the measured wind data and the wind climate predictions are given in terms of the Weibull parameters $A$ and $k$, the mean wind speed and mean power density is calculated by:

$$U_i = \sum_{j=1}^{N_\theta} f_j A_{ij} \Gamma \left( 1 + \frac{1}{k_{ij}} \right) \tag{7}$$

$$E_i = \frac{1}{2} \rho \sum_{j=1}^{N_\theta} f_j A_{ij}^3 \Gamma \left( 1 + \frac{3}{k_{ij}} \right) \tag{8}$$

where $\rho$ is air density (1.225 kg m$^{-3}$) and $\Gamma$ is the gamma function (Troen and Petersen, 1989). Having calculated the wind speed and power density for every possible single-mast prediction, the multi-mast strategies was applied to calculate the final predicted wind statistics:

$$U_P = \frac{\sum_{i=1}^{N_M} W_i U_{Pi}}{\sum_{i=1}^{N_M} W_i} \tag{9}$$

$$E_P = \frac{\sum_{i=1}^{N_M} W_i E_{Pi}}{\sum_{i=1}^{N_M} W_i} \tag{10}$$

where $W_i$ is the mast weight that depends on the chosen strategy (Sec.1.4).

## 2.4  Evaluation method

To evaluate the multi-mast strategies, we compare the measured and predicted wind statistics at each mast position and calculate the average error of all predictions. The "closest mast" results are used as a baseline. The paper does not analyse the baseline
error in any detail; instead, the paper focus on how the multi-mast strategies improve the baseline. The data and model setup used for the "closest mast", "inverse distance weighting" and "most similar" methods are identical, to make an objective comparison of the multi-mast strategies. For each multi-mast strategy, the absolute error of each mast prediction is given by

$$X_i = X_P - X_0 \tag{11}$$

where $X_P$ is either the predicted mean wind speed ($U_P$) or power density ($E_P$) and $X_0$ is the measured value ($U_0$ or $E_0$).
The mean error of a strategy is calculated by

$$\mu = \frac{\sum_{i=1}^{n} X_i}{n} \tag{12}$$





where $n$ is the number of mast predictions. Finally, the standard deviation of the error is given by

$$\sigma = \sqrt{\frac{\sum_{i=1}^{n} (X_i - \mu)^2}{n-1}} \qquad (13)$$

The results shown in the following indicates the mean error, $\mu$, the standard deviation of the errors, $\sigma$, and the number of predictions used to calculate the statistics, $n$.

## 3 Results

### 3.1 All sites

The top row of figure 4 shows histograms of the absolute error of wind speed (left) and power density (right) for each of the 185 mast predictions using "closets mast" as a multi-mast strategy. Gaussian distributions are shown on the histograms, and the mean error and standard deviation of wind speed ($\mu = 0.11, \sigma = 0.63$) and power density ($\mu = 36.3, \sigma = 149$) are given in the legends. The observed mean wind speed and power density of the 185 masts are 7.36 ms$^{-1}$ and 460.7 Wm$^{-2}$ respectively.

The results for "inverse distance weighting" and "most similar" predictor are shown in the middle and bottom row of figure 4. The left column shows that both multi-mast strategies significantly reduce both the mean wind speed error and the standard deviation compared to "closest mast". The standard deviation of the wind speed error is seen to decrease from 0.63 ms$^{-1}$ to 0.54 ms$^{-1}$ for both methods. The right column of the figure shows even more substantial error reductions for power density. The standard deviation of power density decreases from 149 Wm$^{-2}$ to 125 Wm$^{-2}$ for "inverse distance weighting" and from 149 Wm$^{-2}$ to 116 Wm$^{-2}$ for "most similar" mast. Therefore, by selecting the most similar location instead of the closest location, the mean prediction error is significantly reduced (22% for power density).

### 3.2 Dependence on predictors

To clarify the difference between strategies that rely on the representativeness radius (closest mast and inverse distance weighting) and the similarity principle (most similar predictor), this section focus on sites with a specific number of predictor masts. By combining the available dataset, it is possible to generate combinations of sites with 1, 2, 3, 4, 5 or 6 predictors (see Table 2).

Figure 5 shows the histogram of the absolute error of wind speed (left) and power density (right) for the combinations of sites that have seven met masts (6 predictors for each prediction). The histograms of the strategies have different colours, and the legends indicate the mean error and the standard deviation of each. Compared to the "closest mast", the standard deviation of power density decreases from 189 Wm$^{-2}$ to 164 Wm$^{-2}$ (13%) for "inverse distance weighting" and from 189 Wm$^{-2}$ to 124 Wm$^{-2}$ (34%) for "most similar" mast.

The mean absolute error and the standard deviation for site-combinations with 1, 2, 3, 4, 5 and 6 predictor masts are given in Table A1 and Table A2 (Appendix); however, the results are not directly comparable since they are based on different sites.



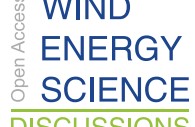

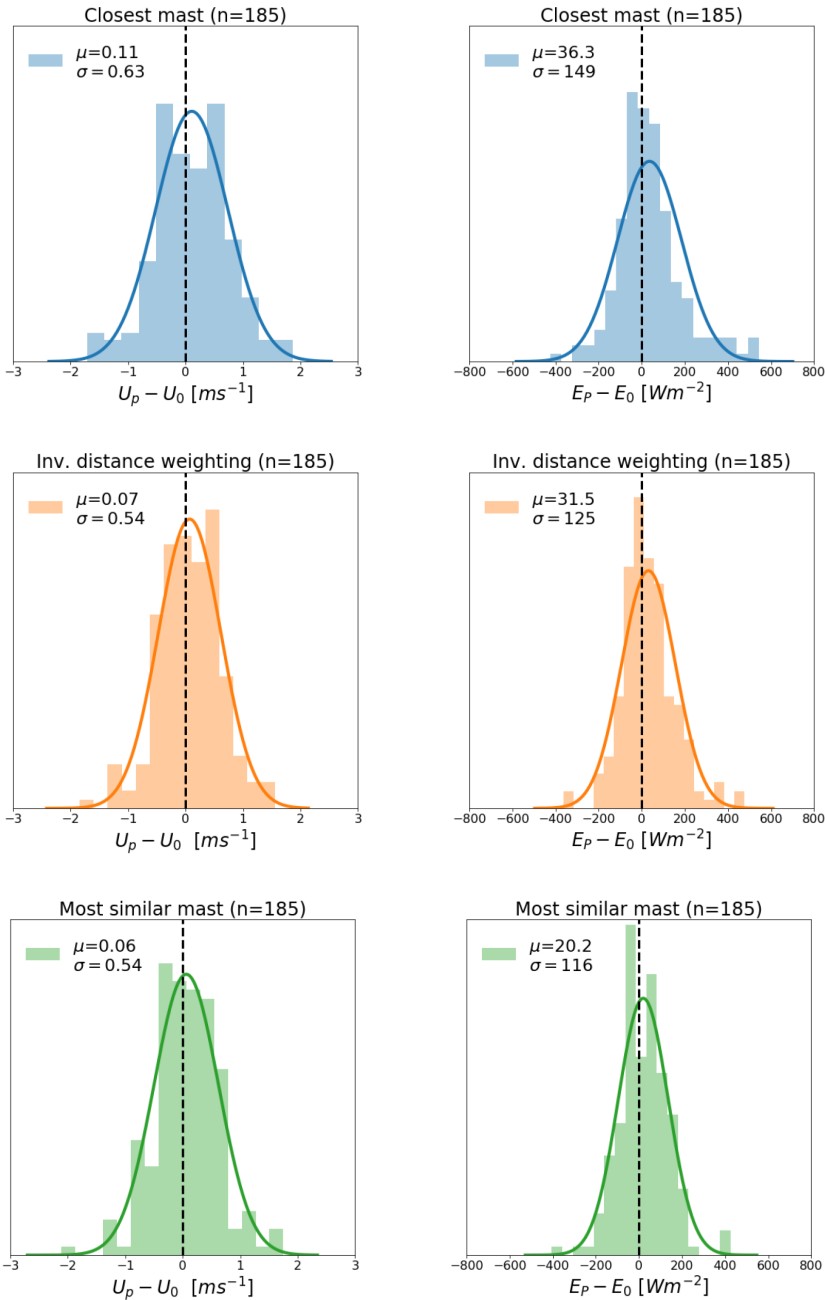

**Figure 4.** Histograms of the absolute wind speed (left) and power density (right) error when using the closest mast (top), inverse distance weighting (middle) and most similar (bottom) strategies. The mean bias and standard deviations are given in the legend. All 185 mast positions have been predicted to make this figure.





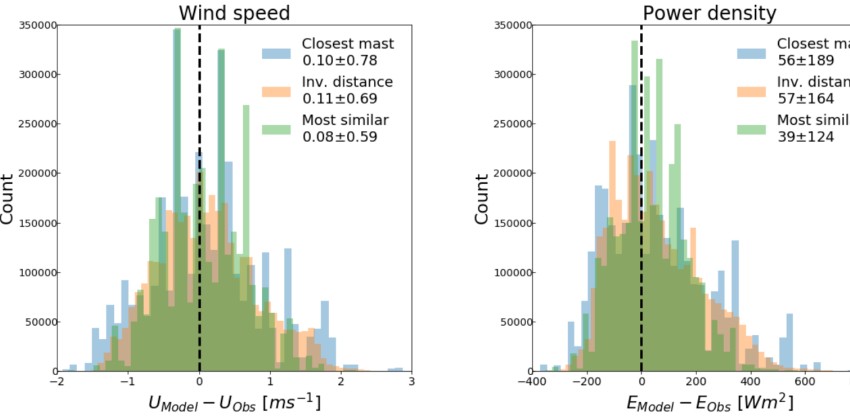

**Figure 5.** Histograms of the absolute wind speed (left) and power density (right) error for sites with 7 masts using the three different multi-mast strategies.

To make a more transparent comparison, we normalise all the results with the results of the "closest mast". Figure 6 shows the average reduction in the standard deviation of wind speed error (left) and power density error (right) for inverse distance weighting and the most similar predictor compared to the closest mast. Note that the multi-mast strategies are the same for sites with only one predictor. The figure shows that both inverse distance weighting and most similar predictor significantly reduce

the average prediction error compared to the closest mast. While inverse distance weighting reduces the error significantly with two predictors compared to one, the added improvement of using three or more predictor masts is much smaller. The most similar strategy has achieved 34% reduction of the standard deviation of power density error with 6 predictor masts, and it appears that the error would keep decreasing with additional predictor masts. It should be noted that the most similar strategy only uses a single predictor mast, but having more options to choose from decreases the error. This indicates that

large improvements could be gained if the "most similar" strategy was used for deciding the location of met masts for resource assessment.

## 4   Conclusions

We have presented a novel method for determining the "most similar" measurement location for wind resource assessment using an expression of the directional-averaged speedup uncertainty. Based on measurements from 185 met masts the "most

similar" met mast is on average a better predictor than the "closest mast" and "inverse distance weighting".

The met masts used in this study have all been positioned by experienced wind and site engineers on well-exposed ridges, and 164 of the 185 met masts have $|\Delta\mathrm{RIX}|$ values below 5%. In the traditional view, this would mean that the closest predictor and the predicted location on average should have similar wind conditions. Despite this, substantial improvements (36% uncertainty reduction on power density predictions) was found by selecting the most similar predictor. This also indicates that




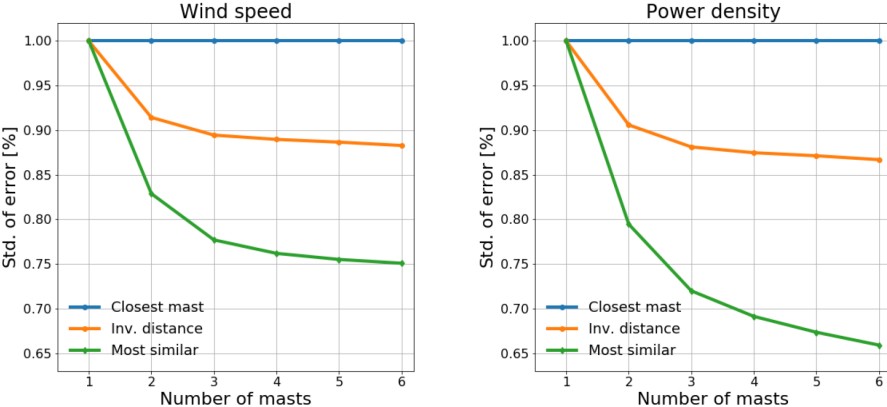

**Figure 6.** Reduction of the standard deviation of wind speed (left) and power density (right) error as function of predictor masts when using inverse-distance weighting and most similar predictor compared to closest mast

even larger error reductions are possible if resource measurement campaigns are designed from the start using the most similar methodology. Designing an optimal measurement campaign using the most similar methodology requires a flow model, a wind turbine layout and an indication of the main wind directions. With these tools, the directional-averaged speedup uncertainty for all wind turbine positions can be calculated and minimised.

The current industry standard is to use inverse distance weighting for resource assessment; the underlying reason is that the

standard error of a weighted mean decrease with the number of independent predictions. Troen and Hansen (2015) demonstrates that the average of two independent flow models does decrease the uncertainty; however, predictions that use the same flow model and measurements from the same time are not independent. The reason why inverse distance weighting improves compared to "closest mast" is probably that it "repairs" a poor choice of predictor mast. Weighted results using inverse "directional-averaged speedup uncertainty" as predictor weight has also been tried (not shown); however, results do not im-

prove compared the most similar predictor. To combine the solution from several masts, it is necessary also to consider the correlation of the errors (Clerc et al., 2012), which is not trivial.

The "most-similar" method is expected to work well for weather models, like the Weather Research and Forecasting (WRF) model, where results have to be interpolated from a calculation grid to the result-point. However, this is a topic for future research.





**Appendix A**

**Table A1.** Wind speed error (mean and standard deviation) depending on the number of predictor masts and selected strategies.

| Masts | 1 | 2 | 3 | 4 | 5 | 6 |
|---|---|---|---|---|---|---|
| Closest mast | $0.09 \pm 0.83$ | $0.09 \pm 0.87$ | $0.09 \pm 0.86$ | $0.09 \pm 0.83$ | $0.10 \pm 0.80$ | $0.10 \pm 0.78$ |
| Inv. distance | $0.09 \pm 0.83$ | $0.10 \pm 0.80$ | $0.10 \pm 0.77$ | $0.11 \pm 0.73$ | $0.11 \pm 0.71$ | $0.11 \pm 0.69$ |
| Most similar | $0.09 \pm 0.83$ | $0.06 \pm 0.72$ | $0.06 \pm 0.66$ | $0.07 \pm 0.63$ | $0.07 \pm 0.60$ | $0.08 \pm 0.59$ |

**Table A2.** Power density error (mean and standard deviation) depending on the number of predictor masts and selected strategies.

| Masts | 1 | 2 | 3 | 4 | 5 | 6 |
|---|---|---|---|---|---|---|
| Closest mast | $46 \pm 194$ | $55 \pm 208$ | $55 \pm 206$ | $55 \pm 199$ | $55 \pm 193$ | $56 \pm 189$ |
| Inv. distance | $46 \pm 194$ | $56 \pm 188$ | $58 \pm 181$ | $58 \pm 174$ | $58 \pm 168$ | $57 \pm 164$ |
| Most similar | $46 \pm 194$ | $40 \pm 165$ | $38 \pm 148$ | $39 \pm 138$ | $39 \pm 130$ | $39 \pm 124$ |

*Data availability.* The data that support the finding of this research is not publicly available due to confidentiality constraints.

*Author contributions.* A. Bechmann analysis; J. P. Murcia Leon analysis asistance; B. T. Olsen paper corrections; Y. V. Hristov wind data and paper corrections.

*Competing interests.* A. Bechmann and B. T. Olsen work in the section at DTU Wind Energy that develop the WAsP software.

*Acknowledgements.* The financial support for the study has been provided by the RECAST project, which is funded by Innovation Fund Denmark (7046-00021B).



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
