# Peer review of "The most similar predictor - on selecting measurement locations for"

_Wind Energy Science, 2020_

## Referee Comment (RC1) · Anonymous Referee #1 · 1 Jul 2020

Thank you for the article, nice work. A few comments: l 21: delete "at" l 30: change: "its distant" to "its more distant" l 42: So is the first part of the equation from Clerck too? if so, write it :) l42: change "wind speed" to "wind speeds" l 53: i think it is $d_i$ not $d_i\hat{}2$ that is the distance? l 55: change "mean decrease" to "mean decreases" l 109: missing a discussion of what the effect of the inevitable differences between the masts have on the final numbers. should they have been weighted according to some quality measure maybe? l118: but the maps could also be of varying quality? l126: find better reference, dont think T&P invented the Gamma function? l135: change "the paper focus" to "the paper focuses" and please donate 10kr to charity every time you make these errors :) l158: add text: compared to closest mast, and 7% compared to inverse

l161: "focuses" 10kr please Figure 4: Maybe a bit of a stretch to show the Gaussian distributions, they dont fit very well! Figure 5: Quite hard to see what message you are trying to convey here. l195: "decreases" 10kr please l204: any thoughts on applications in more complex terrain/bigger differences in RIX?

---

## Referee Comment (RC2) · Anonymous Referee #1 · 7 Jul 2020

thank you for your comments, excellent, i have no further (and thank you for the contribution :) )

---

## Referee Comment (RC3) · Anonymous Referee #2 · 13 Aug 2020

General Comment: The overall structure and flow of this paper is very unconventional and makes it very hard to follow the logic of the paper. I suggest the sections be rewritten with more flow from topic to topic. The introductory material should be made more robust. Perhaps some more background information and references would make the author's thoughts clearer. The Conclusion section is very good, but it was not until the conclusion that the author's thoughts became clear to me. Overall, I think the content is good, but the presentation and writing needs some work.

Specific Comments:

-In the conclusion you acknowledge that this study benefited from having data from

existing met masts. In a real site placement exercise, what spatial resolution do you feel would be necessary to adequately classify a location as "most similar", particularly with respect to terrain? Do you truly feel that WRF, without LES, would be an adequate model to use with spatial resolutions only down to ∼1km?

-Line 88: Define RIX, and how it is determined.

-Section 2.2: Give more details about the model setup. At what resolution are the topography maps? What source did they come from? Are the wind measurements you refer to the data from the 185 masts?

-Where did the met mast data come from? Industry partners? You do not need to specifically name companies, but some indication of the data source should be mentioned. Also, is there any information that can be shared about the anemometer types/ model numbers/ calibration status? Providing more information will speak to the reliability of the dataset used.

---

## Referee Comment (RC4) · Dariush Faghani (Referee) · 18 Aug 2020

Thank you for your contribution. The approach is interesting and the results are promising. Please see below for a few hopefully constructive comments.

1. Have the paper reviewed by a technical writer to correct typos and improve fluency. 2. Lambda and L parameters have been set a priori based on Clerc et al., 2012. Are you suggesting they can be used universally for all sites and flow conditions? 3. Consider using a generic power curve to compare the performance of each approach. 4. The overall tone is very much oriented towards data analysis and processing. It would be good to add more wind resource assessment flavour to it. Especially in

Section 3.2 (Fig. 5) where you focus on just 2 sites. Adding more detail on actual terrain and flow conditions for those sites would be appreciated. 5. In the conclusion you state that your method is "expected to work well for weather models". What is the basis for this claim? If this is just a "teaser" for future work, rephrase appropriately.

Again, thank you for this research and interesting results.

---

## Author Response (AR1)

Thank you for the article, nice work. A few comments: l 21: delete "at" l 30: change: "its distant" to "its more distant" l 42: So is the first part of the equation from Clerck too? if so, write it :) l42: change "wind speed" to "wind speeds" l 53: i think it is $d\_i$ not $d\_i^2$ that is the distance? l 55: change "mean decrease" to "mean decreases" l 109: missing a discussion of what the effect of the inevitable differences between the masts have on the final numbers. should they have been weighted according to some quality measure maybe? l118: but the maps could also be of varying quality? l126: find better reference, dont think T&P invented the Gamma function? l135: change "the paper focus" to "the paper focuses" and please donate 10kr to charity every time you make these errors :) l158: add text: compared to closest mast, and 7% compared to inverse

l161: "focuses" 10kr please Figure 4: Maybe a bit of a stretch to show the Gaussian distributions, they dont fit very well! Figure 5: Quite hard to see what message you are trying to convey here. l195: "decreases" 10kr please l204: any thoughts on applications in more complex terrain/bigger differences in RIX?
* * *
[Figure]

Wind Energ. Sci. Discuss.,
https://doi.org/10.5194/wes-2020-82-AC1, 2020

100 kroner has been sent to doctors without borders for the following errors: l21, l30, l42, l53, l55, l135, l161, l195.

L42: Only Equation 2 is from Clerc; this will be clarified in the manuscript.

L109: It is correct that the results for the different numbers of predictor masts are not directly comparable as they are based on different sites. The results section, there-fore, normalise the "most similar" and the "inverse distance weighting" results with

the "closest mast" to make comparisons possible. The absolute values of the predictions are, however, kept in figure 5 and the appendix for reference. A small discussion/explanation on this should be added.

L118: The maps will be of varying quality, but the different strategies are using the same flow model results irrespectively. So non of the multi-mast strategies have any map-advantages.

l126: Thanks, T&P should not be referenced for the Gamma function, but only on how to apply the function for calculating the wind statistics.

L158: I agree

Figure 4: The fits are weak, but it is useful information that Gaussian distributions does not fit well. I would like to keep the distributions but write that we are aware of the poor fit.

Figure 5: The figure is an example/documentation of the absolute values. The figure is needed to prove that the achieved results are not due to the normalisation. The text needs clarification (also at L109) to highlight how we normalise and where to find the absolute values.

l204: I expect the improvements to be more substantial for more complex terrain, especially for flow models such as WAsP but also for CFD models. The reason for having relatively low dRIX sites in this paper is to not give "most similar" an unfair advantage. WAsP is known to have a bias for high dRIX values and "most similar" will by its nature find masts with low dRIX values (small speedups). Also, the uncertainty on wind speed prediction depends directly on the speedup. By reducing the speedup between predictor and prediction point, all else equal, the uncertainty should decrease.
* * *
[Figure]

Wind Energ. Sci. Discuss.,
https://doi.org/10.5194/wes-2020-82-RC3, 2020

[Figure]

General Comment: The overall structure and flow of this paper is very unconventional and makes it very hard to follow the logic of the paper. I suggest the sections be rewritten with more flow from topic to topic. The introductory material should be made more robust. Perhaps some more background information and references would make the author's thoughts clearer. The Conclusion section is very good, but it was not until the conclusion that the author's thoughts became clear to me. Overall, I think the content is good, but the presentation and writing needs some work.

Specific Comments:

-In the conclusion you acknowledge that this study benefited from having data from

existing met masts. In a real site placement exercise, what spatial resolution do you feel would be necessary to adequately classify a location as "most similar", particularly with respect to terrain? Do you truly feel that WRF, without LES, would be an adequate model to use with spatial resolutions only down to ∼1km?

-Line 88: Define RIX, and how it is determined.

-Section 2.2: Give more details about the model setup. At what resolution are the topography maps? What source did they come from? Are the wind measurements you refer to the data from the 185 masts?

-Where did the met mast data come from? Industry partners? You do not need to specifically name companies, but some indication of the data source should be mentioned. Also, is there any information that can be shared about the anemometer types/ model numbers/ calibration status? Providing more information will speak to the reliability of the dataset used.

————————————————

[Figure]

Wind Energ. Sci. Discuss.,
https://doi.org/10.5194/wes-2020-82-AC2, 2020

[Figure]

I understand the comments about the "flow" of the paper and making the introduction more robust. The introduction has been revisited and expanded with a paragraph that tries to link to the conclusion better. The conclusion has also been edited to try and establish a flow throughout the paper.

Regarding the specific comments:

[Figure]

Q: What spatial resolution would be necessary? A: To answer this question I think we need to clarify that we are operating with two different data sources and therefore two different resolutions: 1. the resolution of the met masts and WRF model results and 2. the resolution of the micro-scale model used to determine the most similar location. Met masts are placed with a spacing of a few km (3.1 km on average in this work); which is similar to the grid spacing of WRF simulations (1-5 km). The resolution of met masts and WRF grid points is therefore comparable. To determine the most similar predictor (mast location or WRF-point), we use a micro-scale flow model that operates with a much finer resolution (1-50 m). A small description is added to the conclusions of the manuscript to clarify this.

Q: Line 88: Define RIX, and how it is determined. A: RIX is defined by the percentage fraction of the terrain along the prevailing wind direction, which is over a critical slope of 0.3. The definition has been added to the manuscript.

Q: Section 2.2: Give more details about the model setup. At what resolution are the topography maps? What source did they come from? Where did the met mast data come from? Providing more information will speak to the reliability of the dataset used. A: I fully understand and agree that transparency regarding the dataset is essential. Confidentiality constraints restrict us in some sense, but the large volume of data also makes it hard to describe it in detail as every wind farm site is different, including anemometer types/calibration and map resolutions. Vestas has provided both the wind and topography data, and we use it directly without corrections or quality control. However, as all data has previously been analysed and scrutinised in connection with wind farm development, we consider the data to have an industry-standard quality. This information is added to the manuscript.
* * *
[Figure]

Wind Energ. Sci. Discuss.,
https://doi.org/10.5194/wes-2020-82-RC4, 2020

[Figure]

Thank you for your contribution. The approach is interesting and the results are promising. Please see below for a few hopefully constructive comments.

1. Have the paper reviewed by a technical writer to correct typos and improve fluency. 2. Lambda and L parameters have been set a priori based on Clerc et al., 2012. Are you suggesting they can be used universally for all sites and flow conditions? 3. Consider using a generic power curve to compare the performance of each approach. 4. The overall tone is very much oriented towards data analysis and processing. It would be good to add more wind resource assessment flavour to it. Especially in

[Figure]

Section 3.2 (Fig. 5) where you focus on just 2 sites. Adding more detail on actual terrain and flow conditions for those sites would be appreciated. 5. In the conclusion you state that your method is "expected to work well for weather models". What is the basis for this claim? If this is just a "teaser" for future work, rephrase appropriately.

Again, thank you for this research and interesting results.

[Figure]

Wind Energ. Sci. Discuss.,
https://doi.org/10.5194/wes-2020-82-AC3, 2020

[Figure]

Thank you for the comments; they help improve the manuscript, and I appreciate it highly.

Q: 1.  Have the paper reviewed by a technical writer to correct typos and improve fluency

A: The fluency of the paper has been improved in the revised manuscript

Q: 2.  Lambda and L parameters have been set a priori based on Clerc et al., 2012.

[Figure]

Are you suggesting they can be used universally for all sites and flow conditions?

A: Lambda and L have not been re-calibrated in this work, but are kept to their original values. In Clerc et al., 2012, the expression is used for estimating energy yield uncertainty for which an accurate calibration may be necessary. For the application of determining the most similar predictor, I believe and suggest, that a precise calibration is less critical.

Q: 3. Consider using a generic power curve to compare the performance of each approach.

A: This is a good idea. However, as the prediction heights and wind speeds vary between the sites, choosing a single power curve representative for all sites is not trivial. The error on the predicted energy yield will depend significantly on the rated power and rated wind speed of the chosen generator.

Q: 4. The overall tone is very much oriented towards data analysis and processing. It would be good to add more wind resource assessment flavour to it. Especially in Section 3.2 (Fig. 5) where you focus on just 2 sites.

A: Fig 5 is the results of more than 33 million predictions, but the text is not very clear on this and has been revised. We are trying to keep the tone towards data analysis, to emphasis that the flow modelling and data is identical for all simulations. The paper attempts to demonstrate that it is a minimal change in the statistical post-processing of results that leads to considerable improvements.

Q: 5. In the conclusion, you state that your method is "expected to work well for weather models". What is the basis for this claim? If this is just a "teaser" for future work, rephrase appropriately.

A: The statement has been rephrased, also following referee #2. The statement was meant as a new hypothesis to prove; a challenge. Choosing the most similar mast location and choosing the most similar WRF result point seems quite comparable. When

downscaling WRF results and interpolating them from the WRF-calculation grid to a result-point, it is common to use the nearest WRF-point or to make a distance-weighted average using the four surrounding grid-points. This is very similar to how resource assessment is done with met masts. I would not be surprised if the "most similar" WRF point were a better approach.

———————————————————

[Figure]

The most similar predictor – on selecting measurem
Bechmann et al.
2020

**Summary of Comments on Marked-up Bechmann et al. - 2020 - The most similar predictor – on selecting measurem.pdf**

**Page: 1**
* * *
**Author: Text Replaced**     Subject: Text     Date: Indeterminate

[Old]: "conducting wind resource assessments with multiple wind measurements and for the optimal design of measurement campaigns."

[New]: "selecting optimal measurement positions for wind resource assessment."
* * *
**Author: Text Replaced**     Subject: Text     Date: Indeterminate

[Old]: "location to a new"

[New]: "loca tion to a prediction"
* * *
**Author: Text Inserted**     Subject: Text     Date: Indeterminate

"micro-scale"
* * *
**Author: Text Inserted**     Subject: Text     Date: Indeterminate

"possible"
* * *
**Author: Text Inserted**     Subject: Text     Date: Indeterminate

"single"
* * *
**Author: Text Inserted**     Subject: Text     Date: Indeterminate

"Much research focuses on the development of improved micro-scale models; however, this work focus on optimal met masts positioning and on how this improves predictions irrespectively of the chosen micro-scale model. This introduction will first explain that met masts today are positioned to minimise the distance to wind turbine positions. We then present a hypothesis that states that met masts should instead be placed following the "similarity principle" and finally, 20 the section describes how the hypothesis will be tested. 1.1 The representativeness radius"

**Page: 2**

Author: Text Deleted    Subject: Text    Date: Indeterminate
"at"

Author: Text Inserted    Subject: Text    Date: Indeterminate
"more"
* * *
**Author: Text Replaced**    Subject: Text    Date: Indeterminate

[Old]: "35 Instead of using distance as the parameter representing similarity, we propose to use the directional-averaged speedup uncertainty, $\sigma_S$ :"

[New]: "parameter that represents similarity, we define the directional-averaged speedup uncertainty, $\sigma_S$ ,"

Font "NimbusSanL-Regu" changed to "NimbusRomNo9L-Regu".
Font-size "8.51801" changed to "9.9626".
* * *
**Author: Text Inserted**    Subject: Text    Date: Indeterminate

"where $f_j$ is the wind direction frequency at the met mast, $N_\theta$ is the number of wind direction sectors and $\sigma_{S2j}$ is the speedup uncertainty of the micro-scale model. As an expression for the speedup uncertainty for a particular wind direction, we use the model by"
* * *
**Author: Annotation Inserted**    Subject: Annotation    Date: Indeterminate
* * *
**Author: Annotation Inserted**    Subject: Annotation    Date: Indeterminate
* * *
**Author: Text Inserted**    Subject: Text    Date: Indeterminate

"(2012):"
* * *
**Author: Text Inserted**    Subject: Text    Date: Indeterminate

"Clerc et al."
* * *
**Author: Text Replaced**    Subject: Text    Date: Indeterminate

[Old]: "speed respectively. $\sigma_S$ combines into one expression, the uncertainty associated with both extrapolation distance and speedup."

[New]: "speeds respectively. $L_1$ and $\lambda$ have not been re-calibrated in this work, but are kept to their original values. $\sigma_S$ combines into one expression, the uncertainty associated with both extrapolation distance and the wind speed difference. 50"
* * *
**Author: Text Inserted**    Subject: Text    Date: Indeterminate

"micro-scale"
* * *
**Author: Text Replaced**    Subject: Text    Date: Indeterminate

[Old]: "decrease"
[New]: "decreases"
* * *
**Author: Text Replaced**    Subject: Text    Date: Indeterminate

[Old]: "inverse distance weighting is not the optimal way of performing resource assessments. Instead, we suggest to follow the similarity principle and determine the "most similar" mast location (not"

[New]: "met masts should be placed following the "similarity principle" instead of reducing extrapo65 lation distance. The hypothesis is verified by showing that the "most similar" predictor gives lower prediction errors than the"

**Page: 4**
* * *
**Author:** Text Replaced     **Subject:** Text     **Date:** Indeterminate

[Old]: "the nearest mast). To validate 60 this approach, we conduct a large number of predictions, test the different multi-mast strategies and compare the results to measurements. We test the following multi-mast strategies:"

[New]: "closest predictor and inverse distance weighting. In the following, we conduct a large number of predictions using the following multi-mast strategies and compare the results to measurements:       "
* * *
**Author:** Text Replaced     **Subject:** Text     **Date:** Indeterminate

[Old]: "method,"

[New]: "predictor,"
* * *
**Author:** Text Inserted     **Subject:** Text     **Date:** Indeterminate

"wind"
* * *
**Author:** Text Replaced     **Subject:** Text     **Date:** Indeterminate

[Old]: "70 1."

[New]: "1. Wind"

Font "NimbusSanL-Regu" changed to "NimbusRomNo9L-Regu".
Font-size "8.51801" changed to "9.9626".
* * *
**Author:** Text Replaced     **Subject:** Text     **Date:** Indeterminate

[Old]: "Wind"

[New]: "Prediction"
* * *
**Author:** Text Replaced     **Subject:** Text     **Date:** Indeterminate

[Old]: "model setup"

[New]: "flow model setups"
* * *
**Author:** Text Inserted     **Subject:** Text     **Date:** Indeterminate

"wind"
* * *
**Author:** Text Replaced     **Subject:** Text     **Date:** Indeterminate

[Old]: "75 2.1 Measurements A dataset has been collected to validate the multi-mast strategies. The dataset consists of measurements from sites with three or more met masts. By having at least three masts, each mast"

[New]: "2.1 Wind measurements The datasets used in this work has been collected specifically to validate multi-mast strategies. It is obtained through wind project developers worldwide and is considered to cover the full spectrum of the conditions experienced in wind projects. The dataset consists of wind measurements from sites with three or more met masts. By having at least three masts, each mast 85"

Font "NimbusSanL-Regu" changed to "NimbusRomNo9L-Medi".
Font-size "8.51801" changed to "9.9626".
* * *
**Author:** Text Replaced     **Subject:** Text     **Date:** Indeterminate

[Old]: "power"

[New]: "project"

**Page: 5**

Author: Text Deleted        Subject: Text        Date: Indeterminate

"for this study"

Author: Text Inserted        Subject: Text        Date: Indeterminate

"where RIX is defined by the percentage fraction of the terrain along the prevailing wind direction, which is over a critical slope of 0.3"

**Page: 6**

**T** Author: Text Replaced     Subject: Text     Date: Indeterminate

[Old]: "sites"
[New]: "predictions"

**T** Author: Text Inserted     Subject: Text     Date: Indeterminate

"million different predictions with 6 predictor masts. The many combinations are primarily possible due to the site that has 25 met masts."

**T** Author: Text Inserted     Subject: Text     Date: Indeterminate

"The results for different numbers of predictor masts are not directly comparable since they are based on different sites."

**T** Author: Text Replaced     Subject: Text     Date: Indeterminate

[Old]: "Figure 3. Mast height and distance to the closest neighbouring mast for all 185 met. masts."
[New]: "and the "inverse distance weighting" results with the "closest mast" to make comparisons of the multi-mast strategies possible."

Font "NimbusRomNo9L-Medi" changed to "NimbusRomNo9L-Regu".
Font-size "8.9664" changed to "9.9626".

**T** Author: Text Inserted     Subject: Text     Date: Indeterminate

"The results section will, therefore, normalise the "most similar""
* * *
**Author: Text Replaced**   Subject: Text   Date: Indeterminate

[Old]: "The table illustrates the number of predictions possible for combinations of sites with specific number of predictors."
[New]: "Number of predictions possible for different numbers of predictors"
* * *
**Author: Text Replaced**   Subject: Text   Date: Indeterminate

[Old]: "Mast combinations"
[New]: "Predictions"
* * *
**Author: Text Replaced**   Subject: Text   Date: Indeterminate

[Old]: "million different predictions with 6 predictor masts. The many combinations are primarily possible due to the site that has 25 met masts. The results section show how the multi-mast strategies perform with these combinations of multi-mast sites. 2.2 Flow model 110 The"
[New]: "120 2.2 Flow model The micro-scale"

Font "NimbusRomNo9L-Regu" changed to "NimbusSanL-Regu".
Font-size "9.9626" changed to "8.51801".
* * *
**Author: Text Replaced**   Subject: Text   Date: Indeterminate

[Old]: "The topography maps and wind measurements needed were provided to the authors and used directly without conducting any corrections. Also, no ΔRIX-or other corrections have been made to the model 115 results, and no tuning of model parameters to improve results was performed. This study represents a statistical analysis of multi-mast strategies. All simulations are based on identical model setups and using the digitised maps, mast locations, anemometer heights and wind data provided to the authors; only the method used to calculate the"
[New]: "No ΔRIX-or other corrections have been made to the model 125 results, and no tuning of model parameters to improve results was performed. The topography maps used for the flow model were provided to the authors and used directly without corrections or quality control. As all data has previously been analysed to be used for wind farm development, and scrutinized by third party companies, the data are considered to have an industry standard quality. The current work is a statistical analysis of the multi- mast strategies. All simulations are based on identical model setups and using the digitised maps, mast locations, anemometer 130 heights and wind measurements provided to the authors; only the method used to calculate the predicted"
* * *
**Author: Text Replaced**   Subject: Text   Date: Indeterminate

[Old]: "Wind statistics 120"
[New]: "Predicted wind statistics"
* * *
**Author: Annotation Inserted**  Subject: Annotation Date: Indeterminate
* * *
**Author: Annotation Inserted**  Subject: Annotation Date: Indeterminate
* * *
**Author: Text Replaced**   Subject: Text   Date: Indeterminate

[Old]: "by:"
[New]: "following"
* * *
**Author: Text Inserted**   Subject: Text   Date: Indeterminate

"(1989):"
* * *
**Author: Text Inserted**   Subject: Text   Date: Indeterminate

"Troen and Petersen"
* * *
**Author: Text Deleted**   Subject: Text   Date: Indeterminate

"(Troen and Petersen,"
* * *
**Author: Annotation Deleted**  Subject: Annotation Date: Indeterminate
* * *
**Author: Annotation Deleted**  Subject: Annotation Date: Indeterminate
* * *
**Author: Text Replaced**   Subject: Text   Date: Indeterminate

[Old]: "gamma function"
[New]: "Gamma function."

Author: Text Inserted      Subject: Text      Date: Indeterminate

"It was also considered to apply a generic wind turbine power curve to allow calculation of energy yield predictions. However, as the prediction heights and wind speeds vary 145 between the sites, using a single power curve for all predictions was not pursued."

Author: Text Replaced      Subject: Text      Date: Indeterminate

[Old]: "focus"
[New]: "focuses"

Author: Text Attributes Changed      Subject: Text      Date: Indeterminate

Font-style changed.

Author: Text Attributes Changed      Subject: Text      Date: Indeterminate

Font-style changed.

Author: Text Replaced      Subject: Text      Date: Indeterminate

[Old]: "All"
[New]: "Original"

Author: Text Replaced      Subject: Text      Date: Indeterminate

[Old]: "the histograms, and 150"
[New]: "the histograms (despite the poor fit), and"
* * *
**Author: Text Inserted**     Subject: Text     Date: Indeterminate

"Histograms of the absolute wind speed (left) and power density (right) error for different combinations of prediction location and 6 predictor masts using the three different multi-mast strategies. Each histogram consists of more than 33 million predictions. 165"
* * *
**Author: Text Replaced**     Subject: Text     Date: Indeterminate

[Old]: "location instead of the closest location, the mean prediction error is significantly reduced (22% for power density)."
[New]: "location, the mean power density error is reduced by 22% compared to closest mast, and 7% compared to inverse distance weighting."
* * *
**Author: Text Replaced**     Subject: Text     Date: Indeterminate

[Old]: "Dependence on predictors 160"
[New]: "New predictor combinations 175"
* * *
**Author: Text Replaced**     Subject: Text     Date: Indeterminate

[Old]: "focus"
[New]: "focuses"
* * *
**Author: Text Inserted**     Subject: Text     Date: Indeterminate

"new"
* * *
**Author: Text Inserted**     Subject: Text     Date: Indeterminate

"As an example of the absolute prediction errors,"
* * *
**Author: Text Replaced**     Subject: Text     Date: Indeterminate

[Old]: "the"
[New]: "a"
* * *
**Author: Text Replaced**     Subject: Text     Date: Indeterminate

[Old]: "the combinations of 165 sites that have seven met masts (6 predictors for each prediction)."
[New]: "predictions that uses 6 predictors. The available dataset allow 33 million different combinations of a prediction location and 6 predictor masts."
* * *
**Author: Text Deleted**     Subject: Text     Date: Indeterminate

"(13%)"
* * *
**Author: Text Deleted**     Subject: Text     Date: Indeterminate

"(34%)"
* * ** * *
**Author: Text Replaced**      Subject: Text      Date: Indeterminate

[Old]: "site-combinations"
[New]: "predictor-combinations"
* * *
**Author: Text Inserted**      Subject: Text      Date: Indeterminate

"normalised"
* * *
**Author: Text Replaced**      Subject: Text      Date: Indeterminate

[Old]: "are the same for sites with only one predictor. The figure"
[New]: "give 190 identical results for predictions with only one predictor. Figure"
* * *
**Author: Annotation Inserted**    Subject: Annotation Date: Indeterminate
* * *
**Author: Text Replaced**      Subject: Text      Date: Indeterminate

[Old]: "180 large improvements could be"
[New]: "significant improvements are"

Font "NimbusSanL-Regu" changed to "NimbusRomNo9L-Regu".
Font-size "8.51801" changed to "9.9626".
* * *
**Author: Text Replaced**      Subject: Text      Date: Indeterminate

[Old]: "was used for deciding the location of met masts for resource assessment."
[New]: "is followed for placement of met masts, especially for single met mast campaigns."
* * *
**Author: Text Inserted**      Subject: Text      Date: Indeterminate

"This proves the hypothesis that met masts should be positioned according to the "similarity principle" instead of reducing the distance to the wind turbines."

**Page: 12**
* * *
**T** Author: Text Replaced      Subject: Text      Date: Indeterminate

[Old]: "Figure 6. Reduction of the standard deviation of wind speed (left) and power density (right) error as function of predictor masts when using inverse-distance weighting and most similar predictor compared to closest mast 190 even larger error reductions are possible if resource measurement campaigns are designed from the start using the most similar methodology. Designing an optimal measurement campaign using the most similar methodology requires a"

[New]: "even larger error reductions are possible if resource measurement campaigns are designed from the start using the most similar methodology, especially for single met mast campaigns. Designing an optimal measurement campaign using the most similar 210 methodology requires a micro-scale"

Font "NimbusRomNo9L-Medi" changed to "NimbusRomNo9L-Regu".
Font-size "8.9664" changed to "9.9626".
* * *
**T** Author: Text Replaced      Subject: Text      Date: Indeterminate

[Old]: "decrease"
[New]: "decreases"
* * *
**T** Author: Text Replaced      Subject: Text      Date: Indeterminate

[Old]: "time"
[New]: "period"
* * *
**⊖** Author: Text Deleted      Subject: Text      Date: Indeterminate

"also"
* * *
**T** Author: Text Inserted      Subject: Text      Date: Indeterminate

"optimally"
* * *
**T** Author: Text Replaced      Subject: Text      Date: Indeterminate

[Old]: "The "most-similar" method is expected to work well for"
[New]: "The "most similar" predictor is a practical alternative. 220 The "most similar" method could also work well for numerical"
* * *
**T** Author: Text Replaced      Subject: Text      Date: Indeterminate

[Old]: "interpolated from a"
[New]: "downscaled and interpolated from the"
* * *
**T** Author: Text Replaced      Subject: Text      Date: Indeterminate

[revised manuscript text omitted]

---

## Referee Report (RR1)

[referee-annotated manuscript omitted]